



# Organic matter characteristics of a rapidly eroding permafrost cliff in NE Siberia (Lena Delta, Laptev Sea region)

Charlotte Haugk[1,2,3], Loeka L. Jongejans[1,2], Kai Mangelsdorf[4], Matthias Fuchs[1], Olga Ogneva[1,5,6], Juri Palmtag[7], Gesine Mollenhauer[5,6], Paul J. Mann[7], P. Paul Overduin[1], Guido Grosse[1,2], Tina Sanders[8], Robyn E. Tuerena[9], Lutz Schirrmeister[1], Sebastian Wetterich[1,10], Alexander Kizyakov[11], Cornelia Karger[4] and Jens Strauss[1]

[1]Permafrost Research Section, Alfred Wegener Institute, Helmholtz Centre for Polar and Marine Research, Potsdam, 14473 Germany
[2]Institute of Geosciences, University of Potsdam, Potsdam, 14476, Germany
[3]now at: Department of Environmental Science and Analytical Chemistry, Stockholm University, Stockholm, 11418, Sweden
[4]Section of Organic Geochemistry, Helmholtz Centre Potsdam GFZ German Research Centre for Geosciences, Potsdam, 14473, Germany
[5]Marine Geochemistry Section, Alfred Wegener Institute, Helmholtz Centre for Polar and Marine Research, Bremerhaven, 27570, Germany
[6]Faculty of Geosciences, University of Bremen, Bremen, 28359, Germany
[7]Department of Geography and Environmental Sciences, Northumbria University, Newcastle-upon-Tyne, NE1 8ST, UK
[8]Institute for Carbon Cycles, Helmholtz-Zentrum Hereon, Geesthacht, 21502, Germany
[9]Scottish Association for Marine Science, Oban, PA37 1QA, UK
[10]now at: Institute of Geography, Technische Universität Dresden, Dresden, 01069, Germany
[11]Cryolithology and Glaciology Department, Faculty of Geography, Lomonosov Moscow State University, Moscow, 119234, Russia

*Correspondence to*: Loeka L. Jongejans (Loeka.Jongejans@awi.de) and Jens Strauss (Jens.Strauss@awi.de)

**Abstract.** Organic carbon (OC) stored in Arctic permafrost represents one of Earth's largest and most vulnerable terrestrial carbon pools. Amplified climate warming across the Arctic results in widespread permafrost thaw. Permafrost deposits exposed at river cliffs and coasts are particularly susceptible to thawing processes. Accelerating erosion of terrestrial permafrost along shorelines leads to increased transfer of organic matter (OM) to nearshore waters. However, the amount of terrestrial permafrost carbon and nitrogen as well as the OM quality in these deposits are still poorly quantified. Here, we characterise the sources and the quality of OM supplied to the Lena River at a rapidly eroding permafrost river shoreline cliff in the eastern part of the delta (Sobo-Sise Island). Our multi-proxy approach captures bulk elemental, molecular geochemical and carbon isotopic analyses of late Pleistocene Yedoma permafrost and Holocene cover deposits, discontinuously spanning the last ~52 ka. We show that the ancient permafrost exposed in the Sobo-Sise cliff has a high organic carbon content (mean of about 5 wt%).We found that the OM quality, which we define as the intrinsic potential to further transformation, decomposition, and mineralization, is also high as inferred by the lipid biomarker inventory. The oldest sediments stem from Marine Isotope Stage (MIS) 3 interstadial deposits (dated to 52 to 28 cal kyr BP) and is overlaid by Last Glacial MIS 2 (dated to 28 to 15 cal ka BP) and Holocene MIS 1 (dated to 7-0 cal ka BP) deposits. The relatively high average chain length (ACL) index of *n*-alkanes along the cliff profile indicates a predominant contribution of vascular plants to the OM composition. The



elevated ratio of *iso* and *anteiso*-branched FAs relative to long chain (C ≥ 20) *n*-FAs in the interstadial MIS 3 and the interglacial MIS 1 deposits, suggests stronger microbial activity and consequently higher input of bacterial biomass during these climatically warmer periods. The overall high carbon preference index (CPI) and higher plant fatty acid (HPFA) values

as well as high C/N ratios point to a good quality of the preserved OM and thus to a high potential of the OM for decomposition upon thaw. A decrease of HPFA values downwards along the profile probably indicates a relatively stronger OM decomposition in the oldest (MIS 3) deposits of the cliff. The characterisation of OM from eroding permafrost leads to a better assessment of the greenhouse gas potential of the OC released into river and nearshore waters in future, which is important to understand the consequences of a warming climate in Arctic environments on the global carbon cycle.

**1 Introduction**

The terrestrial Arctic is highly vulnerable to climate warming as large areas are underlain by ice-rich permafrost (e.g. Strauss et al., 2021b). Terrestrial permafrost ecosystems are affected by ongoing climate warming with consequences for geomorphological, hydrological and biogeochemical processes from a local to regional scale (IPCC, 2019). Almost twice as much carbon is stored in the permafrost region than what is currently contained in the atmosphere (Hugelius et al., 2014;

Mishra et al., 2021), making permafrost carbon dynamics a globally relevant issue (Grosse et al., 2011; Schuur et al., 2008; Strauss et al., 2021a; Turetsky et al., 2020). Total estimated soil organic carbon (SOC) storage for the permafrost region is ~1100-1600 Gt of which 181 ± 54 Gt are attributed to deep permafrost (below 3 m depth) of the Yedoma region (Hugelius et al., 2014; Strauss et al., 2021b; Strauss et al., 2013). Extensive river networks like the Lena River, especially in their delta zones, carry large nutrient and organic matter (OM) loads to the nearshore zone and onto the Arctic Shelf (Mann et al., 2021,

Sanders et al., 2021). The Arctic river discharge increased significantly in recent decades, transporting organic-rich waters to the nearshore area (Holmes et al., 2012; Holmes et al., 2015). Increased river bank erosion of Arctic rivers following warming during the last decades poses an important mechanism of carbon export from land to water (Zhang et al., 2017, 2021; Fuchs et al., 2020).

Warming throughout the Arctic prolongs the season for permafrost thaw and open ice-free water bodies, resulting in increasing

erosion of ice- and carbon-rich permafrost sediments exposed at coasts (Günther et al., 2013; Jones et al., 2020). Very ice-rich permafrost deposits (i.e. 50-90 vol% ice) such as the late Pleistocene Yedoma Ice Complex (Schirrmeister et al., 2013; Strauss et al., 2017) are particularly vulnerable to rapid thermo-denudation and thermo-erosion processes along river shores (Costard et al., 2014; Kanevskiy et al., 2016; Stettner et al., 2018; Fuchs et al., 2020). Vonk et al. (2013b) showed that Yedoma ice-wedge meltwater can increase the decomposition of OM due to co-metabolizing effects. Another potential impact of the

decomposition of terrestrial OM and discharge with Arctic river water is the change in biochemical properties that may increase ocean acidification and anthropogenic carbon dioxide uptake from the atmosphere (Semiletov et al., 2016). Furthermore, Semiletov et al. (2016) estimated that 57% of the terrestrial organic carbon in the East Siberian Shelf originates from ancient



Pleistocene age permafrost C, such as Yedoma deposits adjacent to river or coastal zones. These studies stress the need to better understand the interactions between thawing permafrost and river and nearshore waters.

The study of fossil biomolecules and other OM characteristics provides insights into the composition and level of OM decomposition and hence can greatly improve estimates on the greenhouse gas potential of thaw-mobilised OM from permafrost deposits (Andersson and Meyers, 2012; Sánchez-García et al., 2014). A few studies have previously focused on molecular biomarkers in northeastern Siberian permafrost deposits (e.g. Zech et al., 2010; Höfle et al., 2013; Strauss et al., 2015; Stapel et al., 2016, Jongejans 2020). In general, the abundance and distribution of *n*-alkanes, which are long-chained,

single bonded hydrocarbons, are used for OM characterization where the chain length of *n*-alkanes indicates OM sources.

In our study, we estimate molecular biomarkers (*n*-alkanes, *n*-fatty acids) and use established biomarker proxies and indices such as the average chain length of *n*-alkanes (ACL), the carbon preference index (CPI), and the higher plant fatty acid (HPFA) index to test whether they mirror OM degree of decomposition and reflect the OM quality in ancient permafrost deposits. Additionally, analyses of the total organic carbon content (TOC), the stable carbon isotope ratios ($\delta^{13}$C of TOC), the total

nitrogen content (TN), and TOC/TN (here referred to as C/N ratios), are applied to our sample set. Hierarchical clustering is used to identify the stratigraphical units along the sample profile based on the major changes in OM composition.

Thus, the OM characteristics of permafrost deposits, rapidly eroding at a cliff site in the eastern Lena Delta, are analysed for the first time for biomarkers. The set of frozen samples was obtained along a 25 m vertical cliff profile with relatively high sampling density of about 1 m covering all exposed cryostratigraphic units. In this study, we aim (1) to characterise the OM

composition of ancient permafrost that accumulated under different climate conditions, (2) to assess the degree of decomposition that the OM already experienced, and (3) to hypothesise, based on the decomposition legacy, the potential of future decomposability and microbial decomposition of the permafrost OM.

## 2 Study Area

The Lena River forms the largest delta in the Arctic covering an area of 29,000 km² (Schneider et al., 2009) and discharges

the second highest freshwater load into the Laptev Sea, with a mean annual discharge of 525 km³/yr (Holmes et al., 2018). It also transports summer 'heat' from the south to the north (Yang et al., 2005). The study area on Sobo-Sise Island (Figure 1a-b) is located in the continuous permafrost zone. The island stretches between the Sardakhskaya and Bykovskaya main channels in the eastern part of the delta. In addition to the modern floodplain, there are three geomorphic units in the delta (Grigoriev, 1993; Schwamborn et al., 2002). While the first unit consists of Holocene floodplains, the second unit consists of late

Pleistocene and Holocene fluvial deposits that are mostly located in the north-western part of the delta and are cut off from the current delta dynamics (Schirrmeister et al., 2011b). The third geomorphological unit consists of erosional remnants of a late Pleistocene accumulation plain with ice-rich Yedoma Ice Complex deposits (Schwamborn et al., 2002; Wetterich et al., 2008). According to a landform classification for Sobo-Sise Island, 43% of the land surface is occupied by Yedoma uplands and





Yedoma slopes, 43% are thermokarst basins with the remaining 14% being thermokarst lakes (Fuchs et al., 2018). The terrain
is affected by thermokarst processes (Nitze and Grosse, 2016) and surface thaw subsidence (Chen et al., 2018).

The distinct surface morphology of Sobo-Sise Island includes Yedoma uplands intersected by thermo-erosional valleys and
thermokarst basins. Syngenetic permafrost formation in polygonal tundra landscapes over long periods in the late Pleistocene
formed thick deposits with large ice wedges that are exposed at the cliff (Schirrmeister et al., 2011b, 2020; Strauss et al., 2015;
Jongejans et al., 2018). Schirrmeister et al. (2011b) attributed parts of the third geomorphological unit in the Lena Delta in the
western and southern parts of the delta to remnants of a Yedoma accumulation plain. This formed during the late Pleistocene
when the Lena River had its delta farther north. Radiocarbon ages corroborated that Yedoma deposits on Sobo-Sise Island
accumulated during the late Pleistocene between about 52 and 15 cal kyr BP. Substantial hiatuses were found at about 36–29
cal kyr BP and at 20 to 17 cal kyr BP, which may be related to fluvial erosion and/or changed discharge patterns of the Lena
River (Wetterich et al., 2020a). Middle to late Holocene ages from 6.36 to 2.5 cal kyr BP were found in the uppermost cover
deposits of the cliff, which is also in agreement with other cover deposits found on top of Yedoma such as on the nearby
Bykovsky Peninsula (Schirrmeister et al., 2002; Grosse et al., 2007).

The Sobo-Sise Yedoma cliff has an average height of 22 m with a maximum height of 27.7 m above the river water level (m
arl) (Fuchs et al., 2020) and is affected by fluvio-thermal erosion. The current average shoreline retreat rate, which was
calculated using satellite data, is 15.7 m/yr (2015-2018), which is remarkably high (Fuchs et al., 2020). The Sobo-Sise Yedoma
cliff (72°32 N, 128°17 E; Figure 1c) extends over 1,660 m in length and is facing north to the Sardakhskaya Channel. Here,
the water discharge amounts to about 8000 m³/s during the summer-low period (Fedorova et al., 2015) and the Lena River is
ice-covered for about 8 months per year between October and May. The river ice thickness reaches up to 2 m. Water depth at
the beginning of the Sardakhskaya Channel (close to Stolp and Sardakh islands) can reach up to 22 m (Fedorova et al., 2015)
and is approximately 11 m in front of the Sobo-Sise Yedoma cliff, allowing for water flow underneath the river ice cover
during the winter months (Fuchs et al., 2020).

## 3 Methods

### 3.1 Fieldwork

The Sobo-Sise Yedoma cliff was sampled in three overlapping vertical sediment profiles (Figure 1b) covering the entire
exposed permafrost section (profile SOB18-01: 24.1 to 15.7 m arl, profile SOB18-03: 18.2 to 10.2 m arl and profile SOB18-
06: 13.4 to 0.9 m arl). Each profile was cryolithologically described (see Wetterich et al., 2018, 2020a) and samples were
collected at 0.5 m intervals by rappelling down on a rope from the top of the cliff. We used an axe and hammer to extract
defined cubes of frozen ground (~20x10x10 cm) from the cliff wall. Samples were collected after cleaning and scraping off
the outermost unfrozen and frozen parts of the cliff wall in order to collect frozen uncontaminated samples. Then, the samples
were lifted upward, cleaned and subsampled for biomarker analysis. In total, we collected 61 sediment samples of which 28
were selected for biomarker analysis at about 1 m intervals covering the entire exposed section. The samples were stored



frozen in pre-combusted glass jars, apart from 9 samples (SOB18-06-09 to SOB18-06-34) which were initially stored in plastic whirl packs, before being transferred in a frozen state to glass containers after transport to the laboratories.

## 3.2 Sedimentological organic matter parameters

Prior to bulk geochemical analyses all samples were freeze dried (Sublimator, ZIRBUS technology), grinded and homogenised
(Fritsch pulverisette 5 planetary mill; 8 min at 360 rotations per minute). Total elemental carbon (TC) and total nitrogen (TN) content of sediment samples in weight percentage (wt%) were measured with a carbon–nitrogen–sulphur analyser (Vario EL III, Elementar) with a detection limit of 0.1 wt% for carbon and nitrogen. Sample below this detection limit were set to 0.05 wt% (half the detection limit) so that the statistics could be calculated. Total organic carbon (TOC) content in weight percentage (wt%) was measured with a TOC analyser (Vario Max C, Elementar). The TOC to TN (C/N) ratio has been used
as a rough first indicator of the degree of OM decomposition with decreasing values indicating proceeding decomposition (Palmtag et al., 2015). The stable carbon isotope ratio ($\delta^{13}$C) of TOC reflects both the initial contribution from different plant species and plant components, and OM decomposition processes (Gundelwein et al., 2007). Samples for $\delta^{13}$C analyses were treated with hydrochloric acid (20ml, 1.3 molar), heated on a hotplate (97.7°C for 3h) to remove carbonates, and subsequently washed with distilled water. All $\delta^{13}$C samples were measured using a Delta V Advantage isotope ratio MS equipped with a
Flash 2000 Analyser (ThermoFisher Scientific), using helium as a carrier gas. The $\delta^{13}$C ($^{13}$C/$^{12}$C) value is reported in per mille (‰) compared to the standard ratio Vienna Pee Dee Belemnite (VPDB). In addition, the radiocarbon ages of selected plant remains were determined on a MICADAs system. The radiocarbon dating laboratory procedures are given in Mollenhauer et al. (2021). Original data including a Bayesian age-depth model were adopted from Wetterich et al. (2020a).

## 3.3 Lipid biomarker analyses

Lipid biomarkers provide information on a molecular level about the source of OM, the environmental conditions during deposition, and the degree of decomposition. In this study, we focused on *n*-alkanes in the aliphatic OM fraction and *n*-fatty acids in the polar hetero-compound fraction. Changes in their relative abundance can provide indication on the degree of decomposition (Kim et al., 2005) as outlined below. We analysed the *n*-alkane distributions of all 28 samples and selected 13 samples for the analysis of *n*-fatty acids. The selection of the *n*-fatty acids was made to cover the entire profile continuously
(approximately every 2 m).

### 3.3.1 Extraction and fraction separation

Following freeze-drying and grinding, biomarker subsamples were transferred into glass jars. Extraction and separation was conducted according to Schulte et al. (2000) and Strauss et al. (2015). Samples were processed in two batches, each containing 14 samples. We weighed between 8 and 11 g in extraction cell bodies fit for the accelerated solvent extractor (ASE 200
Dionex). Dichloromethane/methanol (ratio 99:1) was used as a solvent mixture for OM extraction. Each sample was held in a static phase (20 min at 75°C and 5 MPa, following 5 min of heating). Dissolved compounds were then further concentrated at



~42°C using a closed-cell concentrator (TurboVap 500 Zymark) and the remaining solvent was evaporated under $N_2$. Afterwards, internal standards were added: 5α-androstane for the aliphatic fraction, ethylpyrene for the aromatic fraction, 5α-androstan-17-one for the NSO (nitrogen-, sulphur- and/or oxygen) neutral polar fraction and erucic acid for the NSO fatty-acid fraction (80 μl each from respective 100 μg/ml standard solutions). Subsequently, an asphaltene precipitation was performed to remove compounds with higher molecular complexity (asphaltenes) by dissolving the extracts in a small amount of dichloromethane and adding a fortyfold excess of *n*-hexane. Precipitated asphaltenes were removed by filtration through a sodium sulphate filled funnel. Subsequently, the *n*-hexane-soluble portion was separated by medium pressure liquid chromatography (MPLC) (Radke et al., 1980) into three fractions of different polarities: aliphatic hydrocarbons, aromatic hydrocarbons and polar hetero compounds (NSO compounds). Finally, the NSO fractions of 13 samples were split into an acid and neutral polar (alcohol) fraction using a KOH impregnated column. While the *n*-fatty acid potassium salts were attached to the silica gel, the neutral polar compounds were eluted with dichloromethane. After remobilising the *n*-fatty acids by protonation of their salts with formic acid, the *n*-fatty acid fraction was obtained with dichloromethane.

### 3.3.2 GC-MS measurements and compound quantification

*n*-Alkanes and *n*-fatty acids were analysed using gas chromatography coupled with a mass spectrometer (GC-MS; GC - Trace GC Ultra and MS - DSQ, both Thermo Fisher Scientific). Prior to the analyses, *n*-fatty acids were methylated with diazomethane. The GC was equipped with a cold injection system operating in the split-less mode. The injector temperature was programmed from 50 to 300 °C at a rate of 10 °C/s. Helium was used as carrier gas with a constant flow of 1 mL/min. After injection, the compounds of interest were separated on an SGE BPX 5 fused-silica capillary column (50 m length, 0.22 mm ID, 0.25 μm film thickness) using the following temperature conditions: initial temperature of 50°C (1 min isothermal), heating rate of 3 °C/min to 310°C, held isothermal for 30 min. The MS operated in the electron impact mode at 70 eV. Full-scan mass spectra were recorded from m/z 50–600 at a scan rate of 2.5 scans/s. Using the software XCalibur (Thermo Fisher Scientific), peaks in the GC-MS run were quantified using the internal standards for *n*-alkanes and *n*-fatty-acids. All biomarker concentrations are expressed in μg per gram of dry sediment (μg/gSed) and per gram of TOC (μg/gTOC).

### 3.4 Lipid biomarker indices

### 3.4.1 Average Chain Length

The *n*-alkane average chain length (ACL) is the weighted average number of carbon atoms used for determining OM sources. Long chain odd-numbered *n*-alkanes (> C25) are essential constituents that serve as biomarkers for higher terrestrial plants (Schäfer et al., 2016), whereas shorter chain lengths indicate bryophyte, bacterial or algal origin. A change of the ACL in the long chain range can suggest a change in the terrestrial source biota. We used the equation (Eq. 1) first described by Poynter (1989) and then applied by Schäfer et al. (2016), but with a chain interval from C23 to C33 following Strauss et al. (2015) and Jongejans et al. (2018):





$$\text{ACL} = \frac{\sum i \cdot c_i}{\sum c_i} \qquad \textit{(C=concentration, i=carbon number).} \tag{1}$$

### 3.4.2 Carbon Preference Index

The CPI (carbon preference index) was originally introduced by Bray and Evans (1961) as the ratio of odd over even numbered *n*-alkanes and indicates the level of OM transformation, which decreases with progressing maturation. OM decomposition leading to lower CPI values is a measure of thermal alteration referring to rocks or oils on a geological timescale. However, this ratio, as well as the very similar odd-over-even predominance (OEP) ratio, were previously used in Quaternary permafrost deposits as indicator for OM decomposition (Zech et al., 2009; Strauss et al., 2015; Struck et al., 2020; Jongejans et al., 2020).

Based on these studies, we refer to values over 5 as less degraded OM of high-quality. Eq (2) describes the CPI and was modified after Marzi et al.(1993) using $C_{23-33}$ as a chain length interval.

$$\text{CPI}_{23-33} = \frac{(\sum \text{odd C}_{23-31}) + (\sum \text{odd C}_{25-33})}{2 (\sum \text{even C}_{22-32})} \tag{2}$$

### 3.4.3 Higher Plant Fatty Acids

For each sample, the absolute *n*-fatty acid (FA) concentration was measured and the most abundant homologue's chain length
was identified. In addition, we looked at the share of *iso-* and *anteiso*-branched FAs, which are indicators for microbial biomass (Rilfors et al., 1978; Stapel et al., 2016). Furthermore, we calculated the higher plant fatty acids (HPFA) index, which is the relative amount of the long chain *n*-fatty acids to long chain *n*-alkanes in the sediments. The HPFA was introduced by Strauss et al. (2015) following the principles of the HPA index of Poynter (1989), only with using fatty acids instead of wax alcohols (Eq. 3). The HPFA index reflects the degree of preservation of OM due to the higher lability of *n*-fatty acids in relation to *n*-
alkanes (Canuel and Martens, 1996). The preferential decomposition of fatty acids is due to their functional group leading to a chemical polarisation within the molecule forming an attack point for geochemical or microbiological decomposition and/or decarboxylation (Killops and Killops, 2013). Therefore, a decrease in the HPFA index indicates increased OM decomposition. We use this index for internal comparison where higher values (above the mean) indicate a comparatively higher quality OM.

$$\text{HPFA} = \frac{\sum \text{n-fatty acids C}_{24}\text{C}_{26}\text{C}_{28}}{\sum \text{n-fatty acids C}_{24}\text{C}_{26}\text{C}_{28} + \sum \text{n-alkanes C}_{27}\text{C}_{29}\text{C}_{31}} \tag{3}$$

### 3.5 Data Analysis

In order to identify the stratification along the cliff based on the OM characteristics of the permafrost sediments, the data set was clustered using a constrained agglomerative hierarchical clustering of a distance matrix (*chclust* of the *rioja* package, in R version 4.0.4) (Juggins, 2019). We applied the non-parametric Kruskal-Wallis (>2 groups) test for statistical analyses of the data to compare all major parameters (TOC, C/N, *n*-alkanes, ACL, CPI, short and long *n*-fatty acids, HPFA index and
*iso+anteiso*/long-fatty acids) between the identified clusters. In the results section, we report the *p*-values; the correlation coefficients are reported in Supplementary Table 1. As an additional data set, we added the age-depth modelling by Wetterich



et al. (2020a), which is based on Bayesian age-depth modelling (Blaauw and Christen, 2011) and ranged for all sampled horizons from 2.500 (at 23.7 m arl) to 51.880 cal kyr BP (at 0.9 m arl).

## 4 Results

The uppermost sediments of the cliff consisted of Holocene age sediments (from 24.1 to 22.5 m arl; upper part of SOB18-01) on top of late Pleistocene Yedoma sediments from 22.2 m arl down to the cliff base at the river water level. A detailed cryostratigraphic description is given in Wetterich et al. (2020a). All data presented here will be available upon publication in the data repository PANGAEA (Haugk et al., submitted).

### 4.1 Sedimentological organic matter parameters

TOC content was highest in the topmost sample and ranged from <0.1 wt% (below detection limit, sample SOB18-01-18 at 15.7 m arl) to 25.51 wt% (SOB10-01-01 at 24.1 m arl). Both the minimum and maximum TOC values were found within profile SOB18-01 (Figure 2). The average TOC content was 4.94 wt% (standard deviation (sd): 4.7, n=28) and values decreased from the cliff top downwards with two values higher than 10 wt% at 24.1 m arl (SOB18-01-01; 25.51 wt%) and at 16.2 m arl (SOB18-03-05; 11.31 wt%). TN content had an average of 0.33 wt% (sd: 0.18) and the highest value was also found

at 24.1 m arl (SOB18-01-01; 0.83 wt%) and the lowest at 15.7 m arl (SOB18-01-18; <0.1 wt%). C/N ratios ranged from 7.20 to 30.55 (Figure 2) and displayed, except for the uppermost sample, only little variability over the cliff profile with a mean of 13.24 (sd: 4.23). The $\delta^{13}C$ values ranged from -25.2 ‰ (SOB18-01-12 at 18.7 m arl) to -29.4 ‰ (SOB18-03-05 at 16.2 m arl) and had an average of -27.17 ‰ (sd: 1.08). There was a significant negative correlation between $\delta^{13}C$ and C/N values (R: -0.59 (pearson correlation), p<0.01).

### 4.2 Biomarkers

#### 4.2.1 n-Alkanes

n-Alkanes were detected in the range between $n\text{-}C_{12}$ and $n\text{-}C_{35}$. Absolute n-alkane concentrations ranged from 1 to 172 µg/gTOC (mean: 31, sd: 43) for the short ($C_{12}$ to $C_{20}$) n-alkanes and from 119 to 3192 µg/gTOC (mean: 1050, sd: 897) for the long chain ($C_{21}$ to $C_{33}$) n-alkanes (Figure 3). The relative n-alkane concentration increased in the lower part of the cliff, closer

to the river level. The same is true for the absolute concentration of the short chain (mean: 1 µg/gSed, sd: 1) and long chain n-alkanes (mean: 42 µg/gTOC, sd: 32). The main dominating n-alkane chain length was $n\text{-}C_{27}$ in the lower part of the cliff and alternated between $n\text{-}C_{27}$ and $n\text{-}C_{29}$ in the upper part (Supplementary Figure 1). Four samples were dominated by the $n\text{-}C_{31}$ n-alkane.

The n-alkane based ACL showed variations between 27.1 (SOB18-06-19 at 4.5 m arl) and 29.0 (SOB18-01-06 at 21.7 m arl)

with a mean of 28.0 across the cliff (sd: 0.50; Figure 3). The CPI of n-alkanes ($n\text{-}C_{23}$ to $n\text{-}C_{33}$) ranged from 5.76 (SOB18-06-





15; 6.5 m arl) to 16.29 (SOB18-06-05; 11.5 m arl) with a mean value of 9.89 (sd: 2.79). Below 7 m arl, the CPI significantly decreased.

### 4.2.2 *n*-Fatty acids

We found *n*-FAs with carbon numbers between $C_8$ and $C_{32}$ and a strong even-over-odd carbon number predominance.
Furthermore, hydroxy FAs ($C_6$ to $C_8$), *iso*-branched FAs ($C_{10}$ to $C_{19}$), *anteiso*-branched FAs ($C_{11}$, $C_{12}$, $C_{13}$, $C_{15}$ and $C_{17}$), monounsaturated FAs ($C_{16}$ to $C_{20}$ and $C_{24}$), unsaturated *iso*- and *anteiso*-branched FAs ($C_{17}$), cyclopropyl FAs ($C_{17}$ and $C_{19}$), di- and triunsaturated FAs ($C_{18}$) and phytanoic acid were detected (Supplementary Figure 2, Supplementary Table 2). Concentration of long chain *n*-FAs ($C_{24}$ to- $C_{32}$) ranged from 290 µg/gTOC at 24.1 m arl to 2346 µg/gTOC at 1.4 m arl (mean: 1041 µg/gTOC, sd: 655). The most abundant long chain *n*-FA was *n*-$C_{24}$ for all samples, except at 16.7 m arl (*n*-$C_{26}$) (Figure
2 and Supplementary Figure 2). The mid chain *n*-FA ($C_{21}$ to $C_{23}$) concentration ranged from 121 to 1250 µg/gTOC (mean: 463 µg/gTOC, sd: 314) and was highest at 22.7 m arl. The short chain *n*-FA concentration ($C_8$ to $C_{20}$) ranged from 120 to 968 µg/gTOC (mean: 560 µg/gTOC, sd: 212) and was highest in the bottom sample at 1.4 m arl. Among the short chain FAs, the *n*-$C_{16}$ dominated all samples (Supplementary Figure 2). The *iso*- and *anteiso*-branched FAs were more abundant in the bottom section of the cliff and lowest in the middle section and the ratio of *iso*- and *anteiso*-branched saturated fatty acids ($C_{11}$, $C_{13}$,
$C_{15}$ and $C_{17}$) to long chained ($C_{\geq 20}$) *n*-FAs range from 0.03 to 0.32 (mean 0.13, sd: 0.09; Figure 3 and Supplementary Figure 2). The HPFA index had a mean value of 0.63 (sd: 0.11, n=13), a minimum of 0.45 (SOB18-06-17 at 5.5 m arl) and a maximum of 0.86 (SOB18-01-04 at 22.7 m arl) close to the cliff top. Overall, HPFA values below 16 m arl were slightly lower than in the upper section.

### 4.3 Clustering

We identified three main sub-groups using agglomerative hierarchical clustering (Figure 4a): unit I from 24.1 to 22.7 m arl (n=3), unit II from 21.7 to 16.7 m arl (n=7) and unit III from 16.2 to 1.4 m arl (n=18). Our clustering matched the three cryostratigraphic units as defined by Wetterich et al. (2020a) which further corresponded to MIS 1, MIS 2 and MIS 3 from the top to the bottom. The TOC content (Figure 4b) and C/N ratio were significantly highest in unit I and lowest in unit II ($p < 0.01$ and $p < 0.05$, respectively; Figure 2 and Supplementary Table 1). The short and long chain *n*-alkane concentration, expressed
in µg/gTOC, was higher in unit III, but the differences were only significant for the short chain *n*-alkanes ($p < 0.01$) (Figure 4c). The short, mid and long-chain *n*-fatty acid concentrations expressed in µg/gTOC did not differ significantly between the units. The ACL and CPI values were similar for each unit (Figure 3). The HPFA index was significantly different between the units ($p < 0.05$) with highest values in unit I and lowest values in unit II (Figure 4d). The share of *iso*- and *anteiso*-branched FAs compared to long *n*-FAs was highest in unit III and lowest in unit II (Figure 4e), but the differences were not significant.



## 5 Discussion

### 5.1 Terrestrial organic matter at the interface between permafrost and river

#### 5.1.1 Organic matter source

We found that the *n*-alkane distributions were dominated by the long chain *n*-alkanes (≥ C21) and that short chain *n*-alkanes only play a marginal role (Supplementary Figure 1). The most abundant *n*-alkane homologues in the entire dataset were *n*-$C_{27}$, *n*-$C_{29}$ and *n*-$C_{31}$, which indicates that the OM stemmed from higher land plants (Eglinton and Hamilton, 1967). This is confirmed by the dominance of long chain *n*-FAs ($C_{24}$-$C_{32}$) with a strong even over odd carbon number predominance (Supplementary Figure 2). The $ACL_{23-33}$ varied around a mean of 28 across the cliff and there were no significant differences between the three units. The relatively high ACL across the cliff (Figure 3) further indicates a predominant contribution of vascular plants, which corroborates the pollen record presented by Wetterich et al. (2021). Their results indicated the presence of tundra-steppe vegetation during MIS 3-2, while MIS 1 pollen spectra of the uppermost three samples indicated a shift from tundra-steppe to shrub-tundra vegetation. Occasional warmer-than-today summers were recorded during early MIS 3 as well as the presence of low-centre polygons with favourable (stable) aquatic conditions during the MIS 3. Cooler and drier summer conditions as well as unstable (draining and rewetting phases) aquatic conditions were reconstructed for the MIS 2 (Wetterich et al., 2021). In our study, the elevated ratio of *iso* and *anteiso*-branched FAs relative to longer chain (C ≥ 20) *n*-FAs in unit I (MIS 1) and III (MIS 3) compared to unit II (MIS 2; Figure 4e and Supplementary Figure 2) suggests stronger microbial activity during the warmer MIS 3 and MIS 1 periods (Rilfors et al., 1978; Stapel et al., 2016) and points to a higher input of bacterial biomass during that time. Additionally, we found a significant abundance of short chain FAs especially *n*-$C_{16}$ in all samples (Supplementary Figure 2). However, these FAs are not only common in bacterial but also in eukaryotic microorganisms (Gunstone et al., 2007), and thus represent a mixing signal. Therefore, we focused here on *iso-* and *anteiso* FAs as they are more specific biomarkers for bacterial biomass (Kaneda, 1991).

The source and nature of the OM preserved in permafrost influences both its quantity and quality (Jongejans et al., 2018). TOC, TN, C/N and $\delta^{13}C$ variations result from changes in biomass productivity and/or decomposition, from different OM sources, from changes in depositional conditions influencing OM preservation and from different characteristics of cryosol formation. Generally, enriched $\delta^{13}C$ and low TOC and C/N values, as we found in unit II and III (Figure 2), are typical for Yedoma deposits that formed during cold stages (Schirrmeister, 2012). However, climate variations during the last ice age differentiated into warmer interstadials (e.g. MIS 3) and colder stadial periods (e.g. MIS 2) which climatically triggered changes in vegetation and cryosol formation. At the Sobo-Sise Yedoma cliff, the TOC values were higher during MIS 1 and 3 compared to the last Glacial (MIS 2) deposits, suggesting higher OM accumulation, which was presumably triggered by higher biomass production. The TOC values from the MIS 3 and MIS 2 sediments of the Sobo-Sise Yedoma cliff (<0.10-11.31 wt%, mean: 4.01 wt%) were significantly higher (p<0.01) than those of other Siberian Yedoma sites (<0.1-27 wt%, mean: 3.0 wt%; 17 study sites, 719 samples) but very similar to data from Kurungnakh Island (mean: 3.8 wt %) which is located about 70 km west-south-west from the Sobo-Sise Island in the central Lena River Delta (Strauss et al., 2012, 2020). Likely, the high TOC





values in the Sobo-Sise record are a result of past wetter conditions leading to the formation of peat layers. Comparable to the Kurungnakh Island Yedoma record, the Sobo-Sise Yedoma cliff is characterised by silty sediments with multiple layers

enriched in peat pointing to paleosol formation during permafrost aggradation. The MIS 3 deposits contained rather less decomposed twigs and grass remains as well as single peaty lenses (15-20 cm in diameter) and peat layers (10-20 cm up to 130 cm thick). A similar occurrence of single twig remains (2-4 mm in diameter), dark brown spots, finely dispersed organic remains, and peaty lenses (5-25 cm in diameter) were found in MIS 2 deposits, while MIS 1 deposits contained much more peaty components, i.e. numerous peaty lenses (2-25 cm in diameter), which was reflected in higher TOC values compared to

MIS 3 and MIS 2 deposits (Wetterich et al., 2020a).

At 16.2 m arl, we found a peak in TOC (11.31 wt%, SOB18-03-05) and a simultaneously depleted $\delta^{13}$C value (-29.43 ‰). High TOC and low $\delta^{13}$C values have been found to be indicative for peat accumulation and low decomposition under wetter conditions in a more anaerobic regime (Wetterich et al., 2009; Schirrmeister et al., 2011a; Strauss et al., 2012). These peat layers can form by moss accumulation which is hardly decomposed and/or incorporated soon upon accumulation. From a

biomarker perspective, this sample was not much different regarding the biomolecular composition, indicating a similar organic biomass (Supplementary Figure 2). The higher relative abundance of long chain $n$-FAs compared to the short chain FAs which might, with all uncertainties, rather represent microbial biomass, may point to less microbial decomposition and better preservation of OM at the time of deposition. Considering paleoenvironmental studies from Kurungnakh Island in the central Lena Delta, paleosol formation was intensified by relatively warm and wet summers during the climate optimum of the

interstadial MIS 3 between 40 and 32 kyr BP (Wetterich et al., 2008; Wetterich et al., 2014). Therefore, it is very likely that this layer is a buried paleosol layer containing peaty material.

### 5.1.2 Organic matter quality

OM from different vegetation types was incorporated from the active layer into the permafrost during and after different phases of decomposition. The biogeochemical and biomarker proxies outlined in the previous chapter mainly describe the sources

and composition of permafrost OM. In addition, biomarker ratios provide information on the decomposition level of the OM and with that, the potential 'decomposability' (quality) of the respective permafrost OM upon thaw. The OM assessment of this study via agglomerative clustering found overall high OM quality of the Sobo-Sise Yedoma deposits with high CPI values (mean: 9.89) and higher C/N ratios (mean: 13.24) compared to the other Yedoma deposits such as those on the Buor Khaya Peninsula (Central Laptev Sea) with mean C/N values of about 10 (Strauss et al., 2015). The elevated C/N value in the top

sample of the Sobo-Sise record likely results from the influence of modern plants rooted in the active layer. For the rest of the profile, the C/N ratios were rather uniform. The high CPI in our study is comparable to other Yedoma sites as reported by Strauss et al. (2015) for the Buor Khaya Peninsula (mean 11.6) and Jongejans et al. (2018) for the Baldwin Peninsula (western Alaska; 12.2). At the Sobo-Sise Yedoma cliff, the CPI values scattered around a mean of 9.89 and decreased in the lowermost 7 m of the cliff profile. This could probably indicate a higher level of OM decomposition for the lower cliff part, but can also

be influenced by the vegetation type and species prevailing during early MIS 3 with stagnant water and partly warmer-than-





today summer climate conditions (Wetterich et al., 2021). The HPFA values (0.45-0.86, median: 0.61) are a bit higher compared to Yedoma deposits investigated by Strauss et al. (2015) on Buor Khaya Peninsula (0.15-0.69, median: 0.54). Overall, the HPFA significantly decreased downwards (Figure 3; p<0.05), which suggests that the OM is further decomposed downwards. This fits to the assumption that there was more time for OM decomposition for the lower, older cliff parts of the

paleo active layer. A higher decomposition for the lower cliff part is also supported by the highest ratio of the *iso-* and *anteiso-* branched FAs vs long chain FAs in the MIS 3 deposits and indicating a higher relative amount of microbial biomass and suggesting a higher microbial activity during this warmer interval (Figure 4). As outlined above, three stages of permafrost aggradation on Sobo-Sise Island linked to climatic variability were identified according to Wetterich et al. (2020a). OM preservation during these stages is strongly impacted by the duration of freezing and thawing periods, the associated presence

and absence of oxygen in the soil, the related level of microbial activity, and/or physical protection of the OM by the inorganic matrices (e.g., Fe complexation) (Freeman et al., 2001; Hedges and Keil, 1995; Lützow et al., 2006). As these factors are all closely interlinked, it is almost impossible to decipher the control of these processes on the finally preserved OM biomarker signatures.

## 5.2 Implications

Nitzbon et al. (2020) found that terrestrial permafrost-locked OC will be significantly thaw-affected by 2100, and it could be even up to three-fold (twelve-fold) more under warming scenario RCP4.5 (RCP8.5) compared to previous estimates if including thermokarst-inducing processes. Deep OM as characterised in our study can be released by deep disturbance processes such as thermokarst, thermal-erosion or riverbank erosion. Our findings show that freshly thawed and high quality OM is mobilised with annual erosion rates of 15.7 m/yr (2015-18, on the long-term 9 m/yr) (Fuchs et al., 2020). Furthermore,

we suggest that the very ice-rich cliff wall sections are not exposed to aerobic conditions for very long time periods before being eroded into the Lena River. Thus, aerobic microbial decomposition of the OM at the cliff front is presumably playing only a minor role. Additionally, cliff erosion is mainly driven by thermo-erosion and niche formation at the base of the ice-rich Yedoma cliff resulting in block-failure instead of slow gradual cliff retreat (Fuchs et al., 2020). Accordingly, some of the OM in the cliff may not even become exposed to the air and thaw at all before being eroded into the river. Fuchs et al. (2020)

showed an average loss of 5.2 x $10^6$ kg OC and 0.4 x $10^6$ kg N per year (2015-2018). For the OC flux sourced from permafrost and peat deposits (and in particular from erosive locations like our study site on Sobo-Sise Island), Wild et al. (2019) estimated 0.9 x $10^8$ kg C/yr.

By using a biomarker approach (e.g. *n*-alcohols, *n*-fatty acids, *n*-alkanes) on sub-aquatic sediments, van Dongen et al. (2008) found a greater degree of decomposition of the old terrestrial OM released by the eastern great Arctic rivers, including the

Lena River. Thus, they predicted greater remineralisation rates and release of carbon dioxide and methane. Our biomarker findings of terrestrial permafrost fit well into this scenario. Winterfeld et al. (2015) studied the lignin phenol composition of the Lena River, Lena Delta and Laptev Sea nearshore zones, and proposed that OM decomposition is considerable after permafrost thawing on land and during transport and sedimentation in the water. The present study on the OM origin and the



annual OC erosion rates at the Sobo-Sise Yedoma cliff complements ongoing research on mobilisation of permafrost-locked carbon from late Pleistocene Yedoma deposits, while thermal erosion is a widespread and climate-sensitive phenomenon in the Yedoma domain, covering nearly 500,000 km$^2$ in Siberia and Alaska (Strauss et al., 2021b). This indicates the high potential of thermal erosion for mobilisation and release upon thaw of not only large amounts of carbon but also well-preserved OM into the aquatic system of the Lena Delta and nearshore Laptev Sea and Arctic Ocean areas, which will affect local but likely also regional biogeochemical cycles in the marine realm (Grotheer et al., 2020; Tanski et al., 2021, Mann et al., 2021)

and the shelf seas. Once mobilised and transported into inland waters, permafrost-derived OC can be rapidly used by aquatic microorganisms, increasing OM decomposition in riverine and coastal Arctic waters (Vonk et al., 2013a, 2015; Drake et al., 2015; Mann et al., 2015). In contrast, Bröder et al. (2019) showed that more than half of the carbon transported and deposited on the shelf sea floor likely resists decomposition on a centennial scale, while the rest decays faster, but relatively slowly. In addition, Karlsson et al. 2011 hypothesised that Yedoma OC, associated with mineral-rich matter from coastal erosion, is

ballasted and thus quickly settles to the bottom.

Nevertheless, the input of suspended sediments and OM might cause reduced light conditions (Klein et al., 2021, Polimene et al., in review) leading to a reduced photosynthetic uptake of $CO_2$ (Retamal et al., 2007). The higher $CO_2$ production in the aquatic systems will lead to higher $CO_2$ emission, and might further reduce the aragonite saturation of the water (Anderson et al., 2011; Tank et al., 2012). How the different biochemical processes in near-shore waters will interact with ongoing climate

change is still largely unknown and will be the subject of intense scientific research in future.

## 6 Conclusions

Sedimentological and biogeochemical analyses showed that the sediments exposed at the Sobo-Sise Yedoma cliff contain a high TOC content (mean 5 wt%) and well-preserved OM (C/N mean 13, mean CPI: 10) in comparison to other Yedoma permafrost sites. Our study corroborated the paleoenvironmental data from the Sobo-Sise Yedoma cliff from previous research

which suggested that Yedoma formation during the interstadial MIS 3 and the accumulation of the topmost Holocene deposits (MIS 1) were associated with more microbial activity than during the stadial MIS 2. In addition, our findings suggest that mainly high-quality OM has been freeze-locked perennially into permafrost during the late Pleistocene to Holocene. Biomarker parameters indicate a higher level of OM decomposition for the bottom 7 m of the cliff profile of MIS 3 age and less OM accumulation during MIS 2 in contrast to the warmer MIS 3 and 1 sequences is assumed. At the Sobo-Sise Yedoma cliff,

representing an example of rapidly eroding permafrost shorelines in the Lena Delta, OM with a high decomposition potential is being mobilised from almost all sections of the cliff profile. This material is suggested to rapidly enter the fluvial and probably also the offshore aquatic ecosystem. Thus, OM mobilisation at the Sobo-Sise Yedoma cliff and similarly eroding permafrost sites bear the potential to impact the carbon dynamics, the biogeochemistry and the riverine and near-shore marine ecosystems.



*Data availability.* The data presented in this study were submitted to the PANGAEA (www.pangaea.de) data repository and will be freely available. Cryolithological and geochronological data from the Sobo-Sise Yedoma cliff are available in PANGAEA (Wetterich et al., 2020b; doi:10.1594/PANGAEA.919470).

*Author contributions.* CH and JS designed this study and drafted a first version of the manuscript. CH carried out the lipid
biomarker analyses and interpretation, with help from LJ, CK and KM. SW, LS, AK conducted the sampling and field studies. CH, JS and LJ led the manuscript writing. All co-authors contributed to the manuscript writing process.

*Acknowledgments.* This study was carried out within the CACOON project and the joint Russian-German expeditions Lena 2018 and Sobo-Sise 2018 supported by the Samoylov Research Station. We thank Michael Fritz (AWI Potsdam) and Aleksey
Aksenov (Arctic and Antarctic Research Institute St. Petersburg) for supporting the permafrost sampling on Sobo-Sise in August 2018. We thank Justin Lindemann (AWI Potsdam) and Anke Sobotta (German Research Centre for Geoscience) as well as Hanno Meyer and Mikaela Weiner (AWI-ISOLAB Facility) for support with laboratory analysis.

*Financial support.* This study is part of the 'Changing Arctic Carbon cycle in the cOastal Ocean Near-shore (CACOON)'
project, which is co-funded by the German Federal Ministry of Education and Research and by UKRI NERC (BMBF #03F0806A and NERC NE/R012806/1). SW was supported by the German Science Foundation (DFG grant no. WE4390/7-1).

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

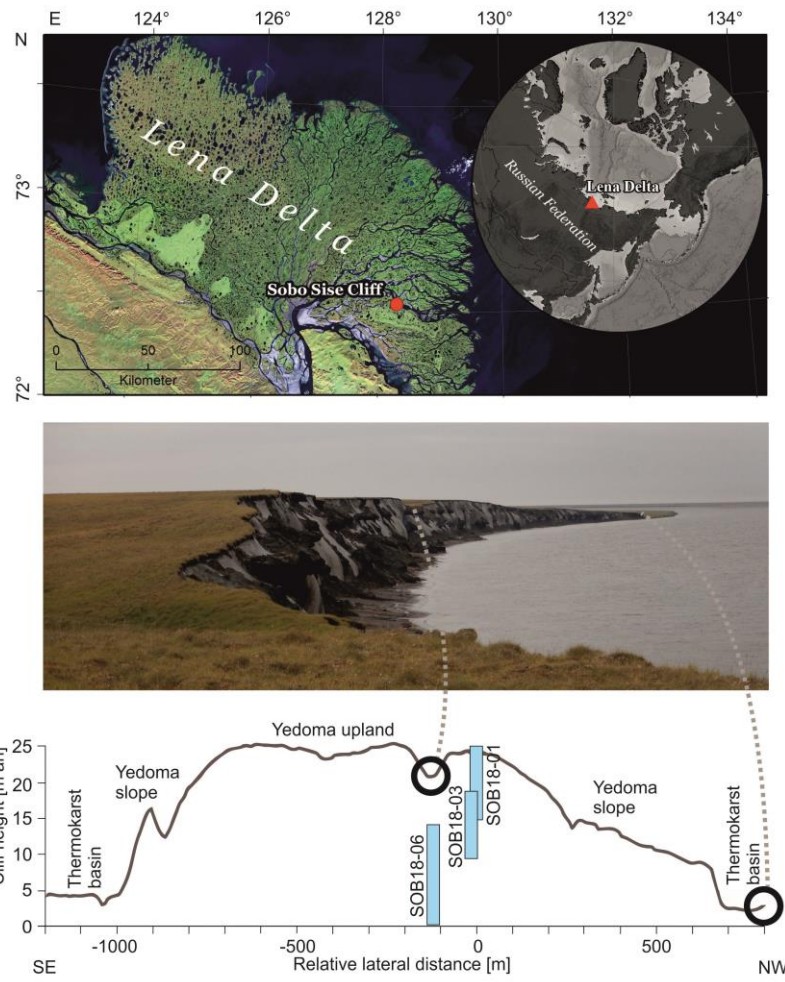

**Figure 1: Overview of the Sobo-Sise Yedoma cliff. (a) Location of the Sobo-Sise Yedoma cliff in the Lena Delta in north-eastern Siberia, Landsat-5 mosaic (band combination 5, 4, 3) including scenes from 2009 and 2010; Landsat-5 image courtesy of the U.S. Geological Survey); (b) picture of the Sobo-Sise Yedoma cliff from the east to west; (c) cross-section of the cliff profile indicating the 705 three vertically sampled sections: SOB18-01, SOB18-03 and SOB18-06, adapted from Wetterich et al. (2020a).**





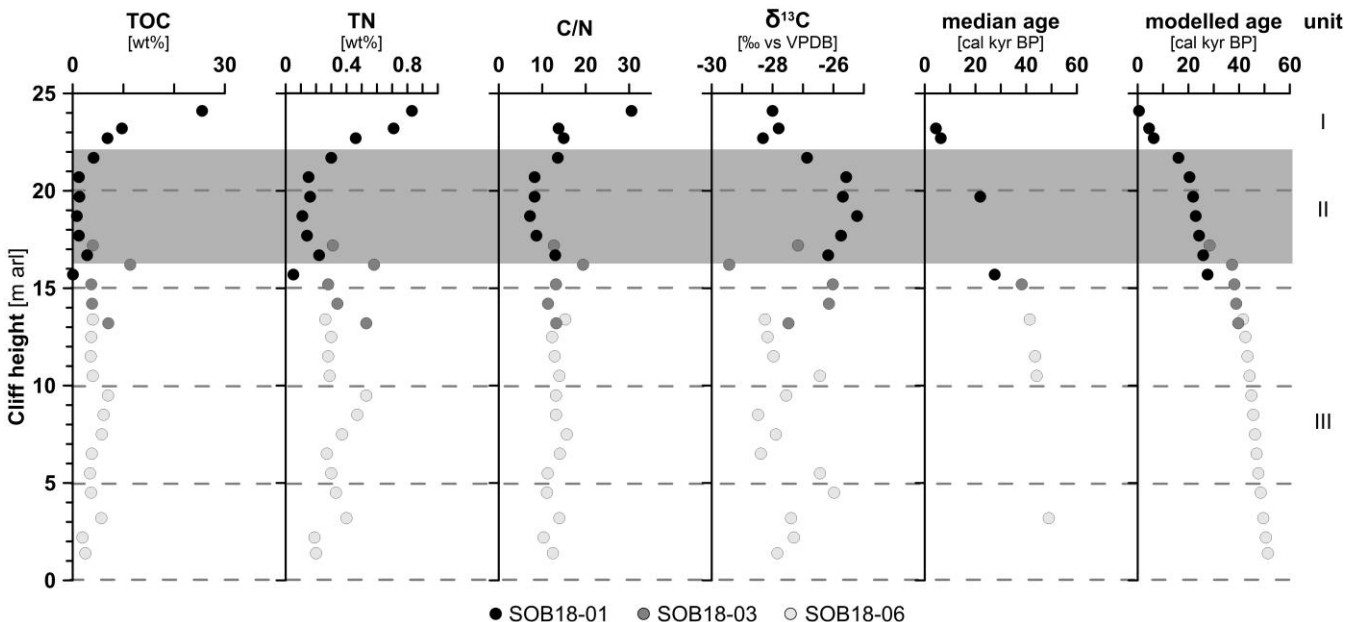

**Figure 2: Biogeochemical parameters of the Sobo-Sise Yedoma cliff: total organic carbon (TOC) content, total nitrogen (TN) content, carbon over nitrogen (C/N) ratio, bulk stable carbon isotope ratios ($\delta^{13}$C), radiocarbon ages and modelled age in calibrated years before present (cal kyr BP). Data points are displayed over cliff height from cliff top at 25 m above river level (arl) to cliff bottom at**
**0 m arl. The three sections of SOB18 are plotted separately for each parameter (black, dark grey and light grey circles). Units I, II (grey rectangle) and III correspond to Marine Isotope Stage (MIS) 1 to 3, respectively. The radiocarbon ages were published in Wetterich et al., 2020a).**

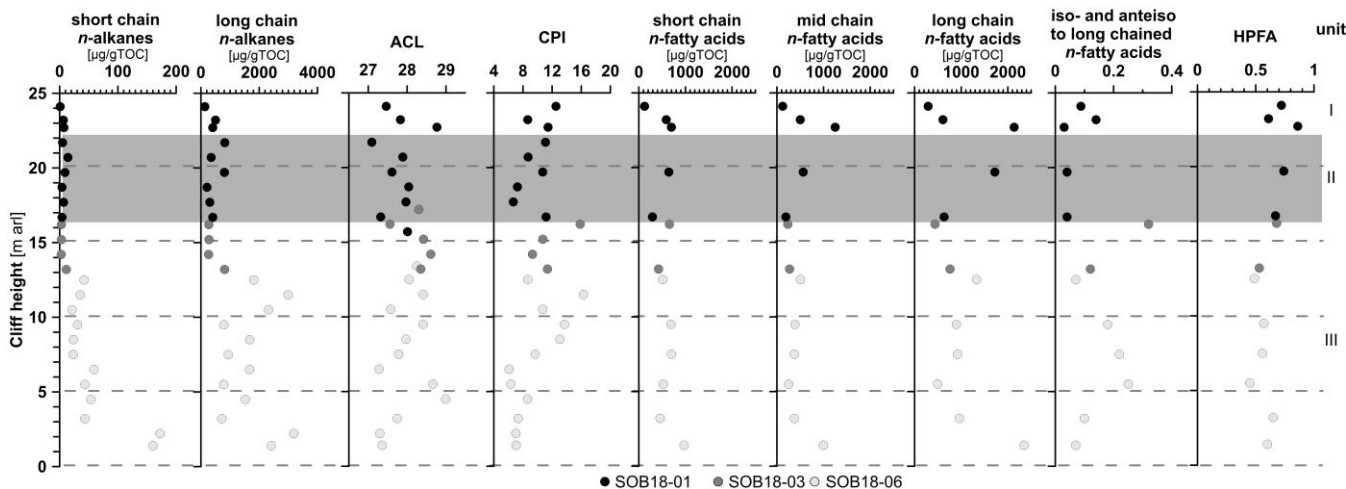

**Figure 3: Biomarker parameters of the Sobo-Sise Yedoma cliff. First four columns: *n*-Alkane parameters, like short chain (C$_{14}$-C$_{20}$,**
**concentration for microbial and algal production), and long chain (C$_{21}$-C$_{33}$, for higher land plants) *n*-alkane concentrations, both in µg/gTOC. After that the *n*-alkane average chain length (ACL$_{23-33}$) and *n*-alkane carbon preference index (CPI$_{23-33}$) is shown. Last five columns: n-fatty acid based parameters, like short (C$_{8}$-C$_{20}$, microbial estimator), mid chain (C$_{21}$-C$_{23}$, estimation for different inputs, for instance from Sphagnum moos species) long (C$_{24}$-C$_{32}$, higher land plants) *n*-fatty acid concentrations in µg/gTOC. After that the ratio of *iso-* and *anteiso*-branched saturated fatty acids (C$_{11}$, C$_{13}$, C$_{15}$ and C$_{17}$) to long chained (C$_{\geq20}$) n-fatty acids is**
**illustrated. The last column shows the higher plant fatty acid (HPFA) index. Data points are displayed over cliff height from cliff**



**top at 25 m above river level (arl) to cliff bottom at 0 m arl. The three sections of SOB18 are plotted separately for each parameter (black, dark grey and light grey circles). Units I, II (grey rectangle) and III correspond to Marine Isotope Stage (MIS) 1 to 3.**

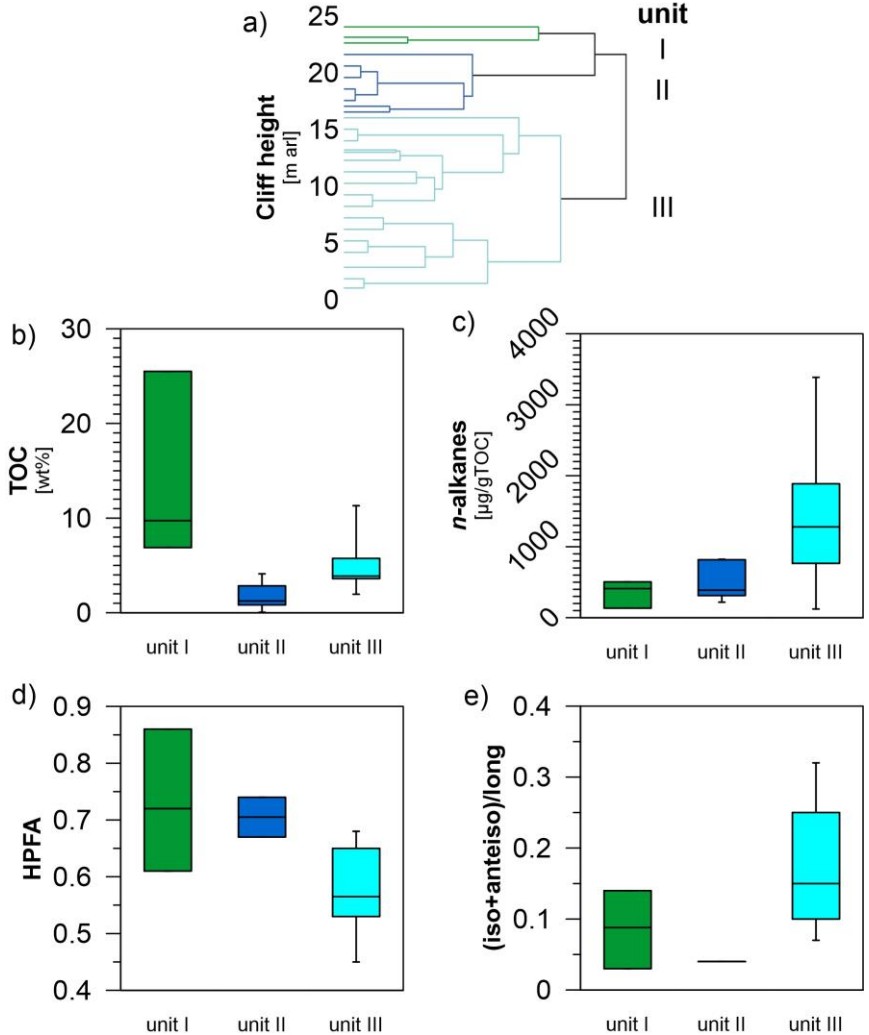

**Figure 4: Statistical separation of the Sobo-Sise Yedoma cliff profile and selected carbon parameters of the separated cliff units; (a) Clustering of samples with y-axis representing cliff height from cliff top at 25 m above river level (arl) to cliff bottom at 0 m arl. Unit I corresponds to Marine Isotope Stage (MIS) 1, unit II to MIS 2 and unit III to MIS 3. Resulting box plots allow better visualisation of OM distribution along the Sobo-Sise Yedoma cliff profile from 25 to 0 m arl, (b) total organic carbon (TOC) content in wt%, (c) $n$-alkane concentration in µg/gTOC, (d) higher plant fatty acid (HPFA) index and e) ratio of *iso*- and *anteiso*-branched saturated** 730 **fatty acids ($C_{11}$, $C_{13}$, $C_{15}$ and $C_{17}$) to long chained ($C_{\geq20}$) $n$-fatty acids.**