# Peer review of "Organic matter characteristics of a rapidly eroding permafrost cliff in NE Siberia (Lena Delta, Laptev Sea region)"

_Biogeosciences, 2021_

## Author Response (AR1)

**Response to the editor's decision letter:**

*Dear Dr Strauss et al,*

*Thank you for submitting your manuscript to Biogeosciences and for responding to the comments/suggestions from both reviewers.*

*Both reviewers suggested minor revision of this manuscript, which also agrees with my own review. Your replies appropriately address their suggestions. Therefore, I invite you to submit a revised version of your manuscript which includes the reviewers' suggestions and your responses.*

Thank you very much.

*I think the grain size data would be a useful addition to this manuscript (without the outlier value) and seems to confirm what reviewer 1 suggested, especially concerning long-chain fatty acids, but I would also discuss the n-alkane data. However, this could be kept brief, focused on the main observations.*

Sure, as suggested we included a short paragraph to our papers discussion (see line 375 ff in the track change document). We included the figure as figure S4 to the supplements

*I also agree with reviewer 1 that at least carbon isotope data (and even hydrogen isotopes) would have added valuable information to increase the quality of this manuscript and to help with the interpretations. Certainly also GDGT data because of the reasons outlined of the reviewer, because it adds valuable insight into the composition, source and preservation vs fate of the organic matter that you studied here, but I can understand this is outside of the scope of your study.*

Thank you. Yes, this is definitely very interesting and a great idea, but as the reviewer 1 and you are stating already, this is a great way to go for a follow up paper. To keep the focus and scope of this study, we would leave this out this time.

*Thank you for these edits, and for your support of Biogeosciences.*

*Sincerely,*

*Dr Sebastian Naeher*

*Associate Editor, Biogeosciences*
* * *
The following responses are basing on our responses at the interactive discussion, just slightly adapted for the final editor advice.

**Anonymous Referee #1**

*Accept with minor revision*

*Summary of the paper: Organic matter characteristics of a rapidly eroding permafrost cliff in NE Siberia (Lena Delta, Laptev Sea region)*

*Haugk et al. studied the characteristic of organic matter (biomarkers, bulk parameters) on a permafrost erosional cliff (located at the Lena Delta, Sobo-Sise Island) that they dated and described. A good amount of samples were studied covering the whole cliff at a 0.5 m resolution which allowed the clustering of the cliff into 3 units. These units are characterized by different biomarker ratios and bulk parameters linked to the quality of the organic matter. MIS 1 and 3 deposit seems to have a stronger microbial biomass activity.*

*I really like this study because of the nice description and the clear clustering of the samples into these 3 units. I am missing a bit more of bulk sedimentological analysis, but it's already a lot of data for such a small scale study. I recommend very minor revisions, it would be amazing if mineral surface area could be measured but I know that takes time.*

Thank you for your constructive feedback. We answered all comments below and revised the manuscript accordingly.

*Main comments:*
*- This study is a very well rounded descriptive study. I am missing a bit of comparison with other permafrost coastal erosion sites such as Muostakh Island (Vonk et al., 2012. Nature); or thaw slump on the Peel Plateau in Canada (Bröder et al., 2021. Environmental Research Letter).*

Thank you, we added more discussion in the last chapter including the suggested studies

*- I was surprised to not see any grain-size or mineral surface area data. When looking at preservation of organic matter, grain size and mineral surface area analysis can give a lot of information as organic matter preserve better when associated with the surface of minerals, in particular long chain alkanes and fatty acids are better linked to the minerals than their short-chain counterpart. So I would advise to measure mineral surface areas or grain size (usually negatively correlated to mineral surface area) for these samples and look at biomarker as loadings (ug m2, see Bao et al., 2018 Influence of Hydrodynamic Processes on the Fate of Sedimentary Organic Matter on Continental Margins).*

The grain size distributions were measured and published by Wetterich et al. (2020; doi:10.1594/PANGAEA.919457). Following your advice, we took a closer look at these data. We added this paragraph (line 375ff in the track change document):

"Previous studies showed that mineral-associated OM can make up a substantial fraction of the OM in permafrost soils which protects the OM from decomposition (Dutta et al., 2006; Mueller et al., 2015). We found that the biomarker concentrations were negatively correlated with the mean grain size published by Wetterich et al. (2020b) (Supplementary Figure 4). Especially, the correlation between the grain size and the short ($p<0.01$), mid and long chain n-FA concentrations ($p<0.05$) were significant. This suggests that, even though the decomposition of n-FAs is generally preferred over n-alkanes, the n-FAs might be relatively better protected from OM decomposition upon mobilisation and transport. The stronger correlation between the grain size and the long chain n-alkanes compared to the short chain counterpart could suggest that the latter might be more vulnerable to decomposition."

In addition, we added a figure to the supplement:

[Figure]

Figure S4: Boxplots of mean grain size vs. biomarker concentrations (in µg g$^{-1}$ sed.) of the Sobo-Sise Yedoma Cliff. Upper row: short (left) and long (right) chain *n*-alkane concentrations. Lower row: short (left), mid (middle) and long (right) chain *n*-fatty acid concentrations. Unit I corresponds to Marine Isotope Stage (MIS) 1, unit II to MIS 2 and unit III to MIS 3. Pearson correlation coefficient R and p-value indicated in right corner of graphs. Note: sample SOB18-01-18 at 15.7 m arl was left out here as the mean grain size in this sample was an outlier (225 µm) compared to the average (29 µm).

*Minor comments:*
*-L42-44: The last sentence is quite a stretch for this study, considering that you only look at a small eroding cliff in a small area of the Lena River Delta. I know that an introduction looks good when it ends with a global statement but this one is quite over the scope of the study.*
We removed the second part of the sentence (the global statement).

*-L45: I think that this sentence would make more sense if it was said that climate warming risk to thaw permafrost, hence arctic region underlain with permafrost might change very rapidly.*
**We rephrased the sentence accordingly.**

*-L47: maybe define what is permafrost, we are not all working in these polar regions.*
We now added the definition of permafrost in the introduction:
"Permafrost is ground that stays below 0 °C for two or more consecutive years."

*-L53-58: The end of the paragraph comes a bit out of the blue, I would move it to the study area part or after L69.*

We moved these sentences further down in the introduction as suggested.

*-L114: "remarkably high" could you add other retreat rate to compare with yours?*
We added the following sentence for comparison:
"In comparison, retreat rates were lower for other Yedoma Cliffs such as on the Kurungnakh Island in the central Lena Delta (4.1-6.9 m $yr^{-1}$; Stettner et al., 2018), at the Itkillik exposure in Alaska (11 m $yr^{-1}$; Wetterich et al., 2008) but even higher for the Muostahk Island (29.4 m $yr^{-1}$; Günther et al., 2013) and Cape Mamontov Klyk (21 m $yr^{-1}$; Günther et al., 2015)."

*-L122: Maybe "Material" instead of "Fieldwork"*
We changed the title to "Sample collection".

*-L144: I guess you dried the samples after washing?*
Yes, we added the following information to the text: "the samples were filtered (Whatman Grade GF/B, nominal particle retention of 1.0 µm) after which the residue was dried and ground."

*-L153: Why did you only select 13 samples? Just curious: was it because of low concentration?*
We made a sample selection because of time constraints.

*-L160: is this a volumetric or weight ratio?*
This is a volumetric ratio. We added this information in the text.

*-L189: Please add more references to this statement or use a review. The use of longchain alkanes to trace for higher terrestrial plants has first been proposed by Eglinton and Hamilton, 1967; Eglinton and Eglinton 2008 and since then used a lot. It is diminishing to only cite Schäffer et al., 2016 although a good study. Furthermore, a reference is missing for the use of shorter chain alkane to trace for bacterial biomass.*
We added the references you mentioned and added references (Cranwell, 1984; Rieley et al., 1991; Kuhn et al., 2010) for using shorter chain *n*-alkanes to trace bacterial/algal biomass.

*-L190: precise what you mean by "long chain range", is that changes above 25 or changes toward 25?*
We removed this, as we do not want to introduce a new "long chain" starting value.

*-L193: Why did you choose to start at C23? Is it to include potential moss influence?*
This is because we followed previous studies Ficken et al. (1998), Strauss, Jongejans etc.

*-L214: Similarly why not include higher chain length of FA, such as C30 and C32?*
We did not include the $C_{30}$ and $C_{32}$ FAs to be consistent with Strauss et al. (2015) who introduced the index.

*-L246: should it be 42 µg gSed-1?*
We rephrased the paragraph and made corrections where necessary.

*-L276: Please repeat which chain length you include in your short, mid and long chain fatty acids*
We added the ranges for short ($C_8$-$C_{20}$), mid ($C_{21}$-$C_{23}$), and long chain ($C_{24}$-$C_{32}$) fatty acids in the text.

*-L277-278: If only the short-chain alkane concentration varies between the units, how come the HPFA differs between the units? From which compounds is the variation coming from?*

The $n$-FA $C_{24}$ is significantly different (p=0.015) between the units and thereby "∑ n-fatty acids $C_{24}$ $C_{26}$ $C_{28}$" (p=0.038), so that makes for a significant change for the complete ratio (p=0.045).

*-L283: You define the long chain alkane starting at n-$C_{21}$, whereas before you included starting nC23. Can you make the manuscript homogenous or detail why you choose to change mid-manuscript.*
We changed the long chain alkanes to start from $n$-$C_{21}$ throughout the manuscript.

*-L284-288: It would have been great to obtain compound-specific d13C for the fatty acids and alkane found in those samples, or even better hydrogen isotopes. Then tracking the differences between units to know more precisely how this permafrost was created (if all FA and alkanes originated from the same region …). I am aware that it is not the scope of this manuscript but maybe an idea for later?*
Thank you, yes.

*-L296-297: iso and anteiso FA are historical biomarkers for bacterial activity but have you thought of branched and isoprenoid GDGT? They are typical for Acidobacteria in soils as well as methanogen/methanotrophic Archaea. These biomarker can add more details to the theory of increased bacterial acticily/biomass.*
This is a great idea for future studies.

*-L297: I totally agree, using C16 FA is tempting because of high abundance, but it has such a large range of source that its interpretation without isotopic signature is too ambiguous.*
Thank you.

*-L391-395: I feel like this paragraph is added without much reason. Of course there is a lot of unknown in coastal biogeochemical processes and more studies to be done, nothing new about that. You could take that paragraph out without changing the scope of this manuscript.*
Removed according your suggestion.

*-Figure 3: in the short-chain panel you could use a logarithmic scale to better capture the variations between SOB18-01 and SOB 18-06.*
The suggestion is good, but for consistency we decided to keep the scale like it is. However, we did change figure 4c from the total $n$-alkane concentration to short chain $n$-alkane concentration.

*Typographical corrections:*
*- Always add a space between number and unit except for % and ‰. At least be consistent throughout the manuscript.*
*- Liter is sometimes written "L" or "l". Be consistent "L" is the official SI abbreviation.*
*- BG guidelines indicate that instead of "µg/gTOC", "µg gTOC-1" should be used, please correct throughout the manuscript.*
*- L88: C25 or 25 atoms of carbon instead of "C25"*
*- L191: C23 to C25*

We revised all suggested typographical corrections throughout the manuscript.

To make all changes visible we uploaded a revised version with track changes.
* * *
**Anonymous Referee #2**

*Haugk et al. worked on organic matter in the permafrost samples collected from the Sobo- Sise Yedoma cliff spanning the last 52 ka in the Lena River Delta. They analyzed C and nitrogen content, bulk carbon isotope and biomarkers to characterize the sources and the quality of OM in the permafrost. They discovered the high quality of permafrost OM for potential decomposition and also demonstrated the associations between permafrost OC characteristics and past climate in the spanning record. This study adds to the present chemical composition data pool of permafrost OC from the Arctic, especially from the eroding Arctic river bank rich in yedoma. This is also well-written and I would recommend a publication with minor revision.*
Thank you for the helpful feedback and comments. We answered all comments below and revised the manuscript accordingly.

*Specific comments:*
*Line 86 Aim 2 and aim 3 might be combined as they are basically telling the same thing.*
We decided to keep the aims as they are as aim 2 targets the present state and aim 3 the future behaviour of the organic matter.

*Line 93 The locations of the three geomorphic units is not all clear. Only the second unit is stated in the north-western part.*
We added a bit more information on the distribution of the geomorphic units as suggested. For details and mapping efforts the reader should read the cited literature.
"While the first unit consists of Holocene floodplains and could occur in the whole delta area, the second unit consists of late Pleistocene and Holocene fluvial deposits that are mostly located in the north-western part of the delta and are cut off from the current delta dynamics (Schirrmeister et al., 2011b). The third geomorphological unit consists of erosional remnants of a late Pleistocene accumulation plain with ice-rich Yedoma Ice Complex deposits and is present mainly in the west, south and east of the delta (Schwamborn et al., 2002; Wetterich et al., 2008, Morgenstern et al. 2011)."

*Line 139 What is the precision of TOC and δ13C measurements?*
We added the analytical accuracy of the TOC (0.1 wt%) and d13C measurements (0.15‰) to the text.

*Line 146 The statement about radiocarbon age is misleading as those data are adopted from Wetterich et al., 2020 but not measurement in this study.*
We moved the information about the radiocarbon ages from the methods to the results only and rephrased the sentence so that it is clear that the data and the age-depth model were published by Wetterich et al. (2020a).

*Line 154 What is the reason of using 13 samples for fatty acids analysis and 28 samples for n-alkane?*
We made a selection because of time constraints.

*Line 161 What is temperature of the following 5 min of heating?*
This was unclear, the heating was to get the 75 °C so before the 20 min static phase. We revised this in the text to:
"Each sample was held in a static phase (5 minute heating phase, 20 min at 75 °C and 5 MPa)."

*Line 231 I guess SOB10-01-01 should be SOB18-01-01?*
You are right, we corrected this.

*Line 231 What is the precision of TOC measurement? I am not sure if two decimal points is really needed.*
Following your suggestions, we now report the values with 1 decimal.

*Line 238 It may be better to keep consistency on the number of decimal points for the same data set.*
As answered to the comment above, we followed your suggestion.

*Line 254 "and" should be "showed"?*
We split the sentence and added the wording you suggested.

*Line 279 It should be "lowest values in unit III" for HPFA index.*
We corrected this.

*Line 313 Can the formation of those peat layers in this area be related to the formation of different geomorphic units?*
Not as far as we know. In the sentence following line 313, we tried to build a regional bridge to the Kurungnakh deposits.

*Line 349 Presumably, when the OM is frozen, decomposition should be very limited and may not cause much difference on the decomposition index. And the decreasing trend of HPFA doesn't seems to be very significant. In addition, how would you explain the change of iso and anteiso FA when HPFA shows consistency in unit III?*
Even though it is difficult to see in Figure 3 (it is clearer in Figure 4; in the revised manuscript we now refer to this figure instead of 3), the HPFA was significantly different between the units (p=0.045). Therefore, we argued that the OM decomposition was advanced furthest in this unit. This corresponds nicely to the highest ratio of *iso-* and *anteiso*-FAs compared to the mid and long chain FAs.

*Line 364. "Our findings show that freshly thawed and…15.7 m/yr" may be rephrased as this study did work on the mobilization of permafrost.*
Thanks, we rephrased the sentence making clear that the erosion rates were found by Fuchs et al. and were not part of this study.

*L373, van Dongen et al 2008 argues that greater degree of decomposition was found in the western Eurasian Arctic. Please check again.*
Thanks, this was a typo, we changes "greater" to "lower".

To make all changes visible we uploaded a revised version with track changes.

---

## Author Response (AR2)

*Comments to the author*:

*Dear Dr Strauss et al,*

*Thank you for the revised manuscript where you have included the comments/suggestions from both reviewers and your responses.*

*I am happy with the changes made, so I think this version can be accepted for publication. However, after a second look at the grain size data, I wonder if you need to include a few more details, so I ask you to consider the following additional, very minor comments:*

*1) Consider including the chain-length range of alkanes and fatty acids into Fig S4 or at least into the figure caption?*

*2) Consider splitting n-alkanes into three categories by adding mid-chain n-alkanes (equivalent to fatty acids)?*

*3) Comment on the common observation that higher molecular weight homologues are typically better preserved than short-chain homologues, which, in addition to chain length dominance due to predominant OM sources, would contribute to the pattern that you see in your results. We also know from previous research that organic compounds are enriched in finer material, so this also contributes to what you see here.*

*4) It further looks like Unit 1 has higher fatty acid contributions relative to n-alkanes, whereas you see the opposite in Unit 2. Noting that alkanes are also derived from fatty acid decarboxylation, the OM in Unit 2 could be more degraded?*

*Thank you for these edits and I look forward to the final version.*

*Sincerely,*

*Dr Sebastian Naeher*

Dear Editor,
Thank you for your positive feedback. Here we respond to your comments:

1) We added the chain length range of the n-alkanes and fatty acids to the figure caption as suggested.

2) We will keep the categories we had as we distinguish between different sources between short (e.g. microbial, algal) and long n-alkanes (higher land plants) and short (microbial) and long FAs (higher land plants). Additionally, we chose to make the further distinction in the long FAs as the mid chain FAs C21-C23 could for instance indicate Sphagnum moss species. However, the mid chain FAs are generally low and only increased in two samples when long chain FAs are also increased. Thus, in these samples the mid chain FAs are most likely also contributions from higher land plants. Therefore, we did not focus on the mid chain FAs in the manuscript.

3) Following your suggestion, we adapted the sentence in L362: "On top of the preferred decomposition of short chain n-alkanes over the long counterpart (Elias et al., 2007), the stronger negative correlation (even though not significant) between the grain size and the long chain n-alkanes compared to the short chain counterpart could suggest that the latter might be more vulnerable to decomposition or might reflect the different sources of these biomolecules. While long chain n-alkanes derive from higher land plants and enter the soil by deposition, short chain n-alkanes might contain a significant proportion of microbial biomass, which abundance depend on the availability of appropriate substrates."

4) In unit II, the long chain FAs still prevail over the long chain n-alkanes, which is also expressed by the HPFA index. We think that you could probably mean that an opposite trend exist in unit III, which seem to be obvious from the long chain FAs and n-alkane data in Figure 3 and is also visible by a decrease in the HPFA (ratio between long chain FA and long chain n-alkanes) data. So yes, this could be an indication that there is an advanced level of OM degradation in the deeper part of the cliff (units III). This might be also supported by the fact that the microbial FAs (iso/anteiso FA) are relative to the long chain fatty acid increased in this interval which suggest a higher microbial activity in this interval. We interpreted that in that way already in section 5.1.2 Line 355-364.

Best regards, Jens Strauss on behalf of the author team